# Biodiversity in Urban Green Space: A Bibliometric Review on the Current Research Field and Its Prospects

**DOI:** 10.3390/ijerph191912544

**Published:** 2022-10-01

**Authors:** Xuancheng Zhao, Fengshi Li, Yongzhi Yan, Qing Zhang

**Affiliations:** 1Ministry of Education Key Laboratory of Ecology and Resource Use of the Mongolian Plateau, School of Ecology and Environment, Inner Mongolia University, Hohhot 010021, China; 2Collaborative Innovation Center for Grassland Ecological Security, Hohhot 010021, China

**Keywords:** urban sustainability, ecosystem service, human well-being, biodiversity

## Abstract

Understanding the development process of urban green space and biodiversity conservation strategies in urban green space is vital for sustainable urban development. However, a systematic review of the urban green space biodiversity research is still lacking. We have retrieved 3806 articles in WOS core journals and carried out the bibliometrics analysis through the three related search terms: urban, green space, and biodiversity. We found that: (1) the year 2009 was a changing point, and the number of articles have increased exponentially since 2009. The United States, China, Europe, and Australia are closely linked, and four research centers have formed; (2) all studies can be classified into three research themes: “Pattern of Urban Green Biodiversity”, “Ecological Function of Urban Green Biodiversity”, and “Sustainability of Urban Green Biodiversity”; (3) based on the evolution of keywords, this field is divided into the budding stage (1998–2012) and the development stage (2012–2021). The keywords in the budding stage focus on the diversity of different species, and the keywords in the development stage focus on the ecosystem services, biodiversity protection, and residents’ satisfaction; (4) the future research focus may be in three aspects: studies on green space in the less urbanized area and urban-rural ecotone, the regulation mechanism and cultural services of urban green space, and the rational layout and management of urban green space. This study hopes to provide a reference for future research on urban green space biodiversity and promote the sustainable development of urban green space.

## 1. Introduction

Currently, nearly 55.3% of the world’s population lives in cities. By 2035, the global urban population is expected to account for 62.5% of the total population, and by 2050, nearly 70% of the population will live in cities [1]. The acceleration of urbanization has triggered some environmental problems, such as the heat island effect [2], air pollution, biodiversity loss, damage to human health, etc. [3]. As an important part of the urban ecosystem, urban green space has improved the urban environment and mitigated the negative impact of urbanization to a certain extent [4]. It plays a crucial role in sustainable urban development [3,5]. Therefore, urban green space has gradually become an important field of urban biodiversity research.

Urban green space originally referred to the open space in the city. It was defined as any land without buildings, or of which not more than 1/20 part is covered with buildings in the city. Open space includes gray space and green space. Gray space refers to the impermeable and hard surface areas, such as the concrete surface of urban sapace. Green space refers to the permeable and soft surface areas, such as soil and lawn [6]. According to the ownership of urban green space, it can be divided into private and public green space [7]. According to the functional attributes of urban green space, it can be divided into public green space and production protection land. The former is mainly used to arrange recreational facilities for residents to share, while the latter has the functions of sanitation, isolation, and safety protection. And according to the size of the area, it can also be divided into small, medium, large, and super large urban green space [8]. Most of the classifications generally emphasize the characteristics of urban green space, such as quantity, size, type, availability, and accessibility.

Urban biodiversity plays an important role in maintaining ecosystem functions and enhancing human well-being [9,10], such as improving air quality, reducing noise, repairing polluted soil, providing food and raw materials for residents, etc. In addition, urban green space has been proven to be a refuge for biodiversity [11,12]. Many studies have also discussed the urban biodiversity patterns, so as to provide better suggestions for biodiversity conservation [13,14]. The study on the biodiversity of urban green space spans multiple scales, including global [15,16], state [17], regional scale [18,19], and even in a small area of the city [20,21]. The research subjects differ and involve fish [22], bees [23], butterflies [24,25], birds [26,27], plants [28], and microorganisms [29]. The diversity dimension includes not only species diversity [30,31], but also functional diversity [32,33], genetic diversity [34], and landscape diversity [35].

Scholars have also widely discussed the ecosystem function of urban green space biodiversity [36]. The biodiversity of urban green space significantly affects the green space microenvironment, such as soil carbon storage [37], soil temperature [38], soil moisture [37], green space biomass [39], etc. Urban green space also plays a positive role in improving urban ecosystem functions, such as mitigating the heat island effect [40], and regulating water runoff, etc. [41]. In addition, urban green space can also effectively deal with the adverse effects and risks of climate change and reduce the disaster losses that are caused by extreme weather and climate events.

In addition, many studies have confirmed that urban green space plays a key role in improving human well-being, such as relieving stress and fatigue [42], reducing noise pollution [43], reducing disease occurrence [44], reducing crime rate [45], and improving education quality [46]. It has greatly improved the living quality of residents and their physical and mental health.

By summarizing the articles in a certain field, bibliometrics can reveal the development trend, research themes, and future focus issues in this field [47,48]. In recent years, a lot of studies on urban green space diversity have emerged. There are 156 reviews in total, including 30 reviews focusing on cities, of which only one focuses on the agriculture biodiversity of urban green space. As far as we have seen, a systematic review based on bibliometrics has not been reported. Therefore, based on the WOS database we adopt the bibliometrics method to analyze the research progress of urban green space biodiversity. Our aim is to solve the following two problems: (1) what is the research progress of global urban green space biodiversity in the past few decades, including the number of articles, the different stages, the differences, and cooperation between countries? (2) What is the research focus of urban green space biodiversity in the future? This study hopes to provide a systematic understanding of the biodiversity of urban green space.

## 2. Method

### 2.1. Data Sources

The Web of Science (WOS) is an academic information resource that covers the widest range of disciplines and contains the largest number of papers, which is the most widely used by scientists. This study is based on the WOS core database for literature retrieval. The topic search includes “urban or city or metropolis” and “green space or green space or green area or green coverage or green belt or park or green surface” and “diversity or biodiversity or richness or abundance”. The retrieval time is limited to 31 December 2021, and 3806 articles were retrieved.

### 2.2. Statistical Method

Bibliometrix is a function package for quantitative research of scientometrics and bibliometrics in R Studio software. It has a relatively complete bibliometric analysis process of data import, transformation, data analysis and scientific visualization, including two series of functions: (1) bibliometric basic analysis and analysis index extraction; (2) mining of literature related concepts, knowledge and social structure. We used the bibliometrix function package to conduct a quantitative analysis of 3806 documents that were retrieved from the WOS database, such as document issuance, co-citation, subject evolution, etc.

### 2.3. Bibliometrics Analysis

#### 2.3.1. Descriptive Analysis, Trend, and Change Point Detection

We analyzed the basic information of the 3806 articles. In order to explore the development of the field, we tested the changing trend through a Mann–Kendall test. In order to explore the significant changes in a certain year, we used Pettitt method to test the change point (Conte et al., 2019) [49]. In addition, the distribution, citation, and cooperation of articles in various countries and journals were also counted.

#### 2.3.2. Theme Exploration

In order to explore the main research directions in this field, the co-citation analysis in bibliometrics was used to cluster the articles, explore the main research contents of each cluster, and show the key articles of each cluster. In addition, in order to clarify the evolution of research topics in this field, word cloud was firstly extracted through wordcloud2 function package to identify the frequency of keywords and find the key issues. Secondly, multiple correspondence analysis (MCA) analysis was conducted to determine the relationship between the keywords (Raza, 2020) [50]. Finally, in order to determine the structure and evolution of research topics in each type of paper, we drew a trend topics map, and made a thematic map based on keyword analysis. The trend topics map is based on keywords and constructs evolution. The thematic map establishes the evolution relationship of topics through two measurement indicators. The horizontal axis is the centrality, which represents the degree of connection between a topic and other topics. The vertical axis is density, which represents the development of a theme.

MCA is a method that allows studying the association between two or more qualitative variables. MCA can also be understood as a generalization of correspondence analysis (CA) to the case where there are more than two variables. A series of transformations allows the computing of the coordinates of the categories of the qualitative variables, as well as the coordinates of the observations in a representation space that is optimal for a criterion based on inertia. The percentage of adjusted inertia that corresponds to each axis and the percentage of adjusted inertia that is cumulated over the two axes are displayed on the map.

#### 2.3.3. Stage Judgment

The same keywords often appear in closely related literature, and such keywords are called co-occurrence keywords. Based on co-occurrence keywords, a research field can be used to divide different research stages, generally including four stages: budding stage, development stage, lull stage and maturation stage [51]. We extracted co-occurrence keywords and extracted the top 50% co-occurrence keywords as node keywords and judged the development stage based on the trend.

## 3. Results

### 3.1. Descriptive Analysis

Descriptive analysis mainly involves indicators such as articles, keywords, authors, institutions, cited times, published years, etc., which are mainly divided into performance and evaluation indicators. Performance indicators describe intuitive phenomena, such as the number of articles, the number of citations, etc. Evaluation indicators are intended to quantitatively evaluate the scientific contributions of articles, authors, magazines, institutions, countries, etc., such as collaboration index, ranking of published articles, etc.

A total of 3806 articles were retrieved from the WOS. The earliest article was published in 1998, and the number of published articles increased exponentially with the change point of 2009 (Figure 1). The average number of citations of 3806 articles per year is 2.9 (Table 1), while in 1999, 2002, and 2007, it reached three peaks, 6.3, 5.8, and 6.9, respectively (Figure 2a). A total of 3806 articles came from 113 countries or regions. The United States had the largest number of articles (700), followed by China (371) and the United Kingdom (265) (Figure 2b). The number of citations of literature that were published in the United States was also the highest (20107), followed by the United Kingdom (16152) and Australia (10459) (Figure 2c). Among the 3806 articles, there were 964 journals (Table 1), and the most published journal was Landscape and Urban Planning (255), followed by Urban Forestry and Urban Greening (254) and Urban Ecosystems (196) (Figure 2d). In addition, a total of 11,855 authors appeared in 3806 articles and the number of articles by a single author was 293. The number of articles that were cooperated by multiple authors was 3513, with an average of 3.11 authors per article, and the cooperation index was 3.3 (Table 1). Between 1998 and 2008, there was only a small part of the cooperation and exchanges between Europe and the United States (Figure 3a). Between 2009 and 2021, Europe, the United States, China, and Australia had the closest academic exchanges, among which the United States had the highest cooperation (464), followed by the United Kingdom (350) and Germany (269) (Figure 3b). 

### 3.2. Co-Citation Analysis

The citation of scientific literature shows the inheritance and utilization of scientific knowledge, as well as the connection and development between events in the process of scientific development. Co-citation analysis aims to compare, analyze, and cluster the phenomenon of co-citation in scientific papers by using library science and statistics.

The co-citation analysis of articles can be roughly divided into three clusters (Figure 4). In Cluster I (green), the article named ‘Biodiversity in Cities Needs Space: A Meta-analysis of Factors Determining Intra-urban Biodiversity Variation’ by Beninde J et al., in Ecology Letters in 2015 is a highly cited document in this cluster, which discussed the driving factors of urban green space diversity. This cluster mainly emphasizes the study of urban green space landscape pattern and biodiversity. In Cluster II (red), Mckinney ML published an article named ‘Urbanization as a Major Cause of Biotic Homogenization’ in Biological Conservation in 2006, which is an important node and highly cited document of the cluster. The paper expounds that urbanization will promote biological homogenization, thus reducing the versatility of urban green space. This clustering mainly emphasizes the ecological function of green space. In Cluster III (blue), Fuller RA et al. ‘s article named ‘Psychological Benefits of Greenspace Increase with Biodiversity’ published in Biology Letters in 2007, is the core highly cited document of the cluster, which studies the satisfaction of human psychology in urban green space of different diversity. In this cluster, the impact of urban green space on the health of residents is mainly emphasized.

### 3.3. Thematic Analysis

Thematic analysis clusters different themes, points out focus, shows the evolution process of themes over time, and forecasts the development trend of future themes. A total of 10,009 keywords (Table 1) were counted in all the articles, of which 27 keywords appeared more than 50 times (Figure 5a). Urban, biodiversity, and urban ecology were the three keywords with the highest frequency. According to the relationship between keywords, the keywords are divided into four categories (Figure 5b). The first category (blue) mainly discusses the green space pattern, habitat change, and species richness in urbanization. The second category (red) mainly discusses the diversity of urban green space and its relationship with ecosystem services. The third category of keywords (green) mainly studies the impact of urban green space on resident health and human well-being. The fourth category of keywords (purple) focuses on national parks.

Before 2012, the number of node keywords increased slowly, but it has increased rapidly since 2012 (Figure 6). Therefore, taking 2012 as a dividing point, the field was divided into two stages, namely, the budding stage (1998–2012) and the development stage (2012–2021). Before 2012, the frequency of keywords is relatively low. The main keywords were populations, species assemblies, biological diversity, etc. More studies were conducted on species diversity in urban green space. The research objects included flora, birds, Formicidae, Lepidoptera, Culicidae, and other groups. Since 2012, the frequency of keywords has increased significantly, and some new keywords have appeared. Keywords such as habitat, urbanization, ecosystem services, challenges, biodiversity, and satisfaction have gradually emerged. The research field focused on the ecosystem service of urban green space and its relationship with human well-being (Figure 7).

The blue part is in the first and second quadrants. Vegetation, climate change, dynamics, and other topics are well developed in this part. The red part is in the first quadrant. Urban, health, benefit, and other topics have developed well and are closely related to other topics. In the first and fourth quadrants, the purple part includes topics such as pattern, community, and ecology which are closely related to other topics. Green and orange parts are in the third quadrant, and the themes of meta-analysis and heterogeneity develop slowly and are relatively isolated (Figure 8).

## 4. Discussion

### 4.1. Temporal and Spatial Characteristics of Biodiversity in Urban Green Space

In terms of time, the number of papers showed exponential growth from 2009 (Figure 1). This may be related to a series of global urban conferences around 2009, e.g., the World Urban Forums in 2006 [52], the 2007 World Conference on urban construction, and the 2006 Eurasian International Conference on sustainable urban development. These conferences emphasized that sustainable urban development requires strengthening urban planning and the rational planning of urban green space. In addition, the United Nations launched the Green Economy Plan in 2008, aiming to accelerate the transition to a green economy and achieve sustainable development. Moreover, the research on the biodiversity of urban green space has gone through two stages: the budding stage (1998–2012) and the development stage (2012 to present) (Figure 6). At present, it is still in the development stage, and a lot still has to be done in the future. The research is developing more slowly than the research on the urban heat island effect. Compared with the diversity of urban green space, the research on the urban heat island effect developed earlier and was studied more widely. The reason for the slower development of this research field may be that the surface temperature in the city is more easily perceived by residents and more closely related to human life [53], while the changes of urban green space biodiversity is often not so obvious. Human beings often perceive the change through urban microclimate and temperature change. In terms of location, Europe, the United States, China, and Australia have a large number of publications, high citation rate, and strong cooperation in the research on biodiversity of urban green space, and gradually formed four research centers (Figure 3b). This is consistent with Kamalski and Kirby [47], which also found that the areas with the most urban research were Europe, the United States, China, and Australia, and formed four research areas. Among the four research areas, the United States, Australia, and Europe are all developed countries, and their urbanization has reached a very high level in the early stage [54,55]. As a developing country, China has accelerated its development rapidly in recent years, and urban planning has significantly improved. The proportion of population in urban areas has exceeded that in rural areas, entering the development period of big cities. [1]. Therefore, it has attracted extensive attention from global scholars, especially Chinese scholars, and has become a research center with many knowledge institutions. Although China’s scientific research achievements rank second, the number of citations is small. Considering that China has only gradually developed in recent years, it takes a certain amount of time for the accumulation of scientific research achievements. In addition, Chinese scholars pay more attention to regional studies.

In terms of research objects, in the budding stage, more attention was paid to different biological groups in the city, involving birds, insects, and plants. Then, in the development stage, the research content included the protection of biodiversity, habitat, ecosystem services, and resident satisfaction (Figure 7). This is similar to the results of Lucia Rocchi et al. [48]. They also conducted keyword clustering and theme trend research, and finally found that the keywords in the later stage tend to be related with human well-being. In the early stage, the research focused on some issues that are closely related to the field, such as agricultural landscape, agricultural ecology, etc. In the late stage, it gradually began to study biodiversity, organic agriculture, and human well-being, and gradually moved closer to the interests of mankind.

### 4.2. Three Themes of Biodiversity Research in Urban Green Space

The key topics of one research will vary greatly in different periods [56]. Our research found that the key issues on biodiversity of urban green space mainly focuses on three themes, namely, “pattern of urban green biodiversity”, “ecological function of urban green biodiversity”, and “sustainability of urban green biodiversity”.

The theme of the “pattern of urban green biodiversity” focuses on the diversity of urban green space in different types of green space, different cities, and different biological groups. It is related to biology, urban ecology, urban planning, genetics, environmental science, and other disciplines [57]. The research objects involve many biological groups, such as plants, ants, birds, and microorganisms [58,59]. The theme of the “pattern of urban green biodiversity” has developed rapidly in recent years. It has been studied in most cities around the world and has formed a variety of research methods in different countries and regions [60].

The theme of the “ecological function of urban green biodiversity” highlights the ecosystem function of urban green space, including microclimate regulation, water regulation, pollution reduction, restoration of biological habitat, and so on [61]. The research object is mainly urban green space or dominant plants in urban green space [62]. This theme is mainly based on ecosystem ecology. Control experiments and mathematical modeling are the main methods [63]. The theme of the “ecological function of urban green biodiversity” focuses on the level of urban functionality.

The theme of the “sustainability of urban green biodiversity” focuses more on the relationship between urban green space and sustainable human development. This topic is based on ecosystem services and sustainability science. Through the questionnaire and scenario simulation, such as a questionnaire on residents’ sports activities or physical and mental health, and scenario simulation of the temperature, climate, and other factors, this theme pays more attention to the health and feelings of residents [64]. This theme recognizes the importance of urban green space diversity, and provides scientific guidance for urban green space optimization [65]. In this theme, the ultimate goal of urban green space is to achieve the improvement of human well-being.

### 4.3. Future Research Focus of Urban Green Space Biodiversity

Our analysis shows a large amount of research evidence that is aimed at understanding the development process of urban green biodiversity and urban sustainability. Focus research can be mainly divided into three aspects. The theme of the “pattern of urban green biodiversity” mainly studied the biodiversity of urban green space under different patterns. Research key topics may focus on three aspects in the future. (1) At present, the research on biodiversity of urban green space pays more attention to high urbanization areas and less attention to areas with a low degree of urbanization, such as urban transition zones and towns [66]. In the future, the research on these areas need to be strengthened. (2) Microorganisms, are essential in the ecosystem and play an important role in nutrient decomposition and the energy cycle [67]. However, there is little research on their role in urban green space at present. Therefore, in the future, we need to invest more efforts in the study of microbial groups in the urban green space (Figure 8). (3) The urban green space includes plants, birds, insects, and other biological groups. There is a need to carry out studies on the interactions between multiple groups and different trophic levels to reveal the influence mechanisms among different groups in urban green space. 

The theme of the “ecological function of urban green biodiversity” explored the ecological function of urban green space, and future research may focus on two aspects. On the one hand, more attention has been paid to the relationships between urban green space and urban temperature, pollutants, habitat, and human health in the past [68,69]. However, the regulation mechanisms of water [70], carbon, nitrogen, and phosphorus in green space [71] are also crucial. Due to the uniqueness of the urban ecosystem, there is no scientific framework for the study of nutrient regulation in the urban green space now [72]. Therefore, based on ecosystem ecology, the regulation mechanism of urban green space is a focus issue in the future. On the other hand, cultural services play an important role in cities, but the current research on cultural services of urban green space is little [73]. Climate and living preferences often lead to different cultural characteristics of urban green space [74]. We suspect that the cultural services of urban green space will form a variety of different research frameworks.

The theme of the “sustainability of urban green biodiversity” has been a key topic in recent years. The main directions of future research may include three aspects: (1) A large number of studies have been carried out on housing construction [75], green public space protection [76], cultural heritage [77], etc. However, there are relatively few studies on urban green space irrigation water and residential water, especially in the arid and semi-arid regions [78,79]. (2) Land use change and habitat fragmentation lead to species loss, and urban green space provides a refuge for biodiversity [80]. In the future, the relationship between urban green space patterns and biodiversity will be revealed, so as to provide scientific guidance for the rational planning of urban green space. (3) Climate change will be an important threat to urban sustainability in the future [81]. The biodiversity of urban green space will help to cope with climate change [82]. It is very important to explore the mechanism and strategy of diversity of urban green space to cope with climate change. 

## 5. Conclusions

The biodiversity of urban green space plays a huge role in promoting human welfare and achieving sustainable urban development. We reviewed the field of urban green space biodiversity, hoping to help scholars from all countries understand its theme and prospects. Through bibliometric analysis, we can see that research on urban green space biodiversity has increased significantly with the change point of 2009 and has formed two stages: the budding stage and the development stage. There are three main themes in this field. The theme of “pattern of urban green biodiversity” focuses on the diversity of urban green space. The theme of “ecological function of urban green biodiversity” focuses on the ecosystem functions of urban green space biodiversity. The theme of “sustainability of urban green biodiversity” focuses on urban green space biodiversity and human well-being. Urban green space is built to protect biodiversity and ultimately improve human well-being. In the future, urban green space should be regarded as one of the important components of urban sustainability. We should invest more effort in rational planning according to the construction strategies of different cities. Besides, according to the 17 sustainable development goals that were proposed by the United Nations in 2015, we should solve ecological problems in the future in combination with society and economy, so as to protect global biodiversity, improve the quality of urban ecosystems, improve the health and well-being of urban residents, and finally move towards a sustainable path.

## Figures and Tables

**Figure 1 ijerph-19-12544-f001:**
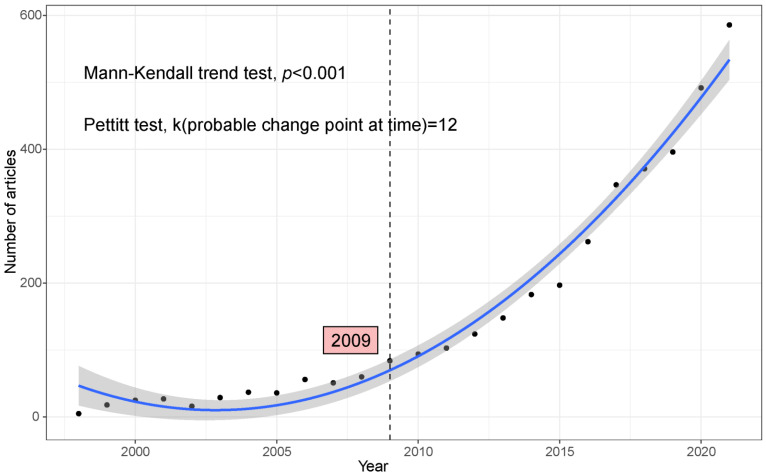
Publication trends and change point of articles based on Mann-Kendall and Pettitt tests.

**Figure 2 ijerph-19-12544-f002:**
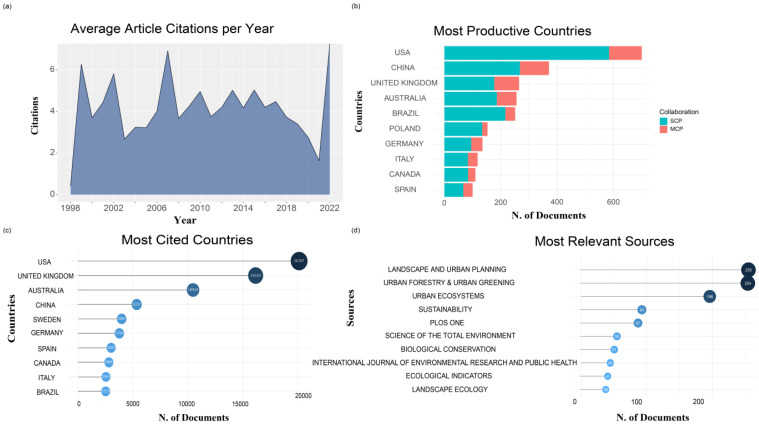
Basic features of all the articles. (**a**) The average number of citations per year from 1998 to 2021. (**b**) Top ten countries in terms of the number of publications, in which SCP stands for single country publications and MCP stands for multiple country publications. (**c**) Top ten countries with in terms of the number of cited articles. (**d**) Top ten journals in terms of the number of publications.

**Figure 3 ijerph-19-12544-f003:**
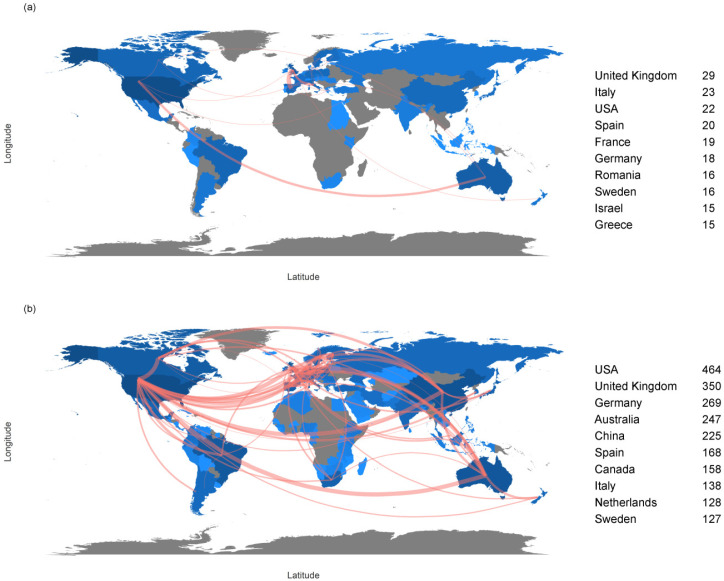
Country collaboration map. The red lines represent the cooperation and exchanges between the authors of the two countries. The more lines, the closer the exchanges. The numbers on the right represent the times a country exchanged scientific products with other countries. (**a**) 1998 to 2008. (**b**) 2009 to 2021.

**Figure 4 ijerph-19-12544-f004:**
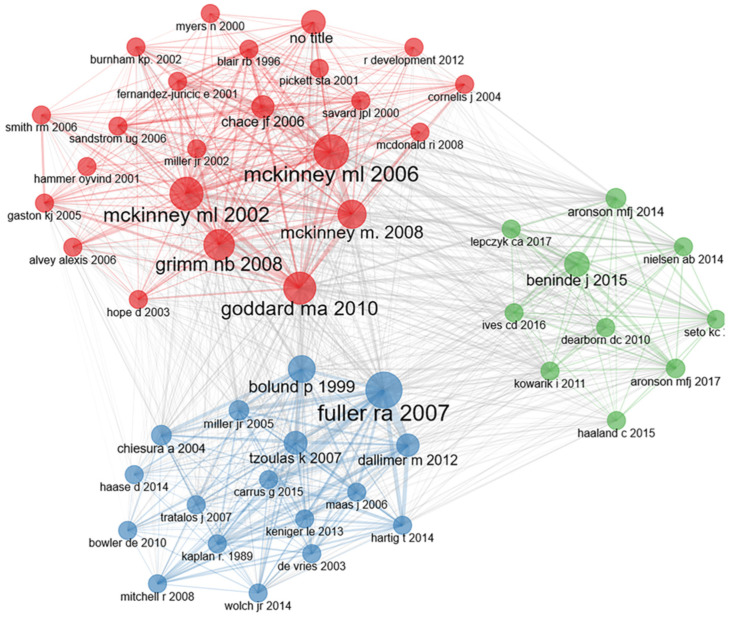
Co-citation network. Green, red, and blue represent the first, second, and third article clusters, respectively.

**Figure 5 ijerph-19-12544-f005:**
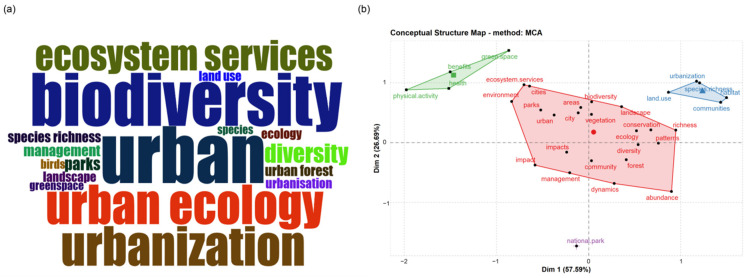
Keywords analysis. (**a**) The word cloud which has 27 keywords with a frequency of over 50 (**b**) According to the keywords, four color clusters of red, green, blue, and purple are formed, and different clusters show different keyword topics.

**Figure 6 ijerph-19-12544-f006:**
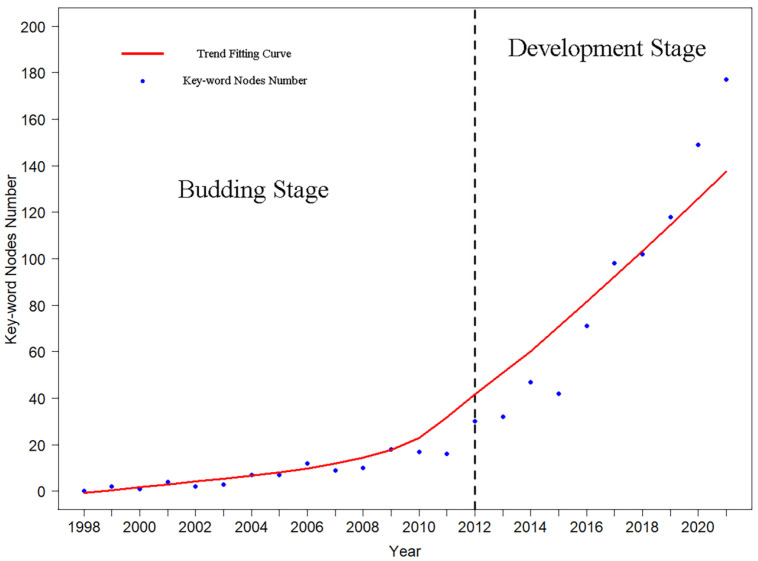
Annual change of keyword nodes number and research development progress.

**Figure 7 ijerph-19-12544-f007:**
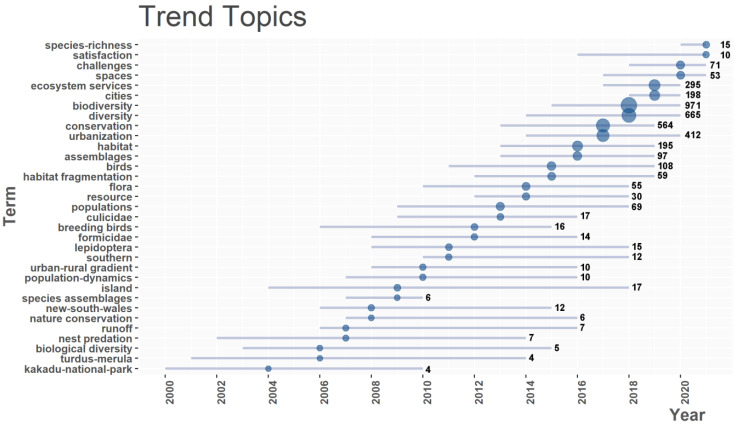
Keywords evolution. The horizontal line in the figure represents the time range when the occurrence frequency of a keyword is greater than four, and the circle represents the highest occurrence frequency in that year. The larger the circle, the higher the corresponding occurrence frequency. The number after each line represents the value in the circle.

**Figure 8 ijerph-19-12544-f008:**
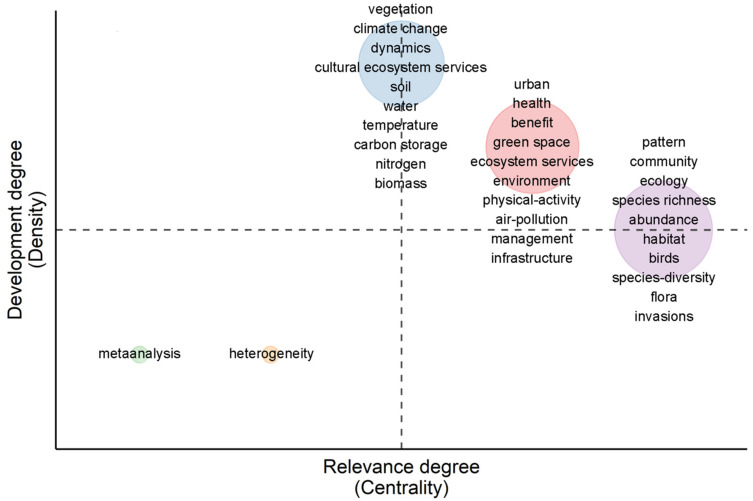
Thematic map. The horizontal axis is the centrality, which represents the relevance degree between a certain theme and other themes. It can be interpreted as the importance of this theme in the entire field development. The vertical axis is the density, which represents the development degree within a certain theme. It can be interpreted as the development status of the theme.

**Table 1 ijerph-19-12544-t001:** Basic features of all the articles.

Description	Results
Number of journals	964
Author’s Keywords	10,009
Average citations per year per articles	2.906
Authors	11,855
Single-authored articles	293
Multi-authored articles	3513
Authors per articles	3.11
Collaboration Index	3.3

## Data Availability

Publicly available datasets were analyzed in this study. This data can be found here: [https://population.un.org/wup/].

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
