# Peer review of "Biodiversity in Urban Green Space: A Bibliometric Review on the Current Research Field and Its Prospects"

_ijerph, 2022, doi:10.3390/ijerph191912544_

Round 1
Reviewer 1 Report
The topic of the review deals with the urban green space as an important element of sustainable urban development. The article is clear and well-organised, but it requires some small improvements that contribute to broadening the topic.
1) When discussing the functions of urban green areas, it would be useful to consider their role in climate change adaptation. It is now an important direction of action in national adaptation strategies and plans (NASs and NAPs).
2) The aim of the article should be rewritten. According to the article’s content it should be rather an attempt to find answers for formulated questions than to solve problems. Additionally, no research problems were indicated.
3) Are the world conferences really the only explanation of the change point in 2008? This is also the time when the term green economy appeared. Green economy as a mean to achieve the sustainable development definitely contributed to increase of researchers interest in studied topic. Additionally, it would be worth to consider whether the publication of 17 SDGs by UN in 2015, including the goal on sustainable cities, had an impact on analysed topics.
4) Conclusions overlap too much with the abstract. They require some short development.
Reviewer 2 Report
Biodiversity in urban green areas is a relevant field of study, on which a systematic review might make an important contribution. For the submitted paper ‘Development and the sustainability of the urban green biodiversity: The current situation and future prospects’ to realize this potential, I believe several improvements are necessary.
a. The title does not seem to represent the approach of the article. Rather than analysing development practice and sustainability, it deals with topics and metrics of the research field. A suggestion: ‘Biodiversity in urban green areas: a bibliographic review on the current research field and its prospects’
b. A convincing problem statement is lacking. Why is it needed to have a better overview of the research field and its prospects, in the academic context and/or in practice? The ‘gap’ statement in line 12 that a systematic review is lacking, does not convince the reader of the urgency, and one wonders whether there are not yet some good overviews available of literature of biodiversity in cities. The article itself already cites a meta-analysis by Beninde (in the latter part), which is a form of systematic review.
c. The described method is a statistical analysis of a list of publications, while especially the future prospects seem to be the result of a qualitative literature review, of which the method is not explained. Recommendation: either choose for the quantitative approach exclusively, or describe in more detail and in order the mixed approach. Why and how are the methods combined?
d. The article contains (too) many figures, several of which are not explicitly used in the text to make a point. The purpose of several figures remains unclear, for a lack of specific research questions. For example: location information is used to affirm the existence of 4 research centers, but what does that contribute to the main issue of knowing what current and future themes are? The focus of the paper seems to be development of themes over time, so why not select figures to emphasize that?
More detailed remarks per line in the manuscript:
15/20 If 2008 is the changing point in the development of the field, why separate budding and development stages in the year 2012 and not in 2008?
38 Why ‘therefore’? The phrase means ‘urban green space has gradually become a hot spot for urban biodiversity’, correct? How does this follow from the above?
41 Why is ‘walled or not’ relevant?
42/45 references in caps, unlike rest of article
48 What is the function of public green space? What is production protection land?
53 Which ecosystem functions, and why?
55 What do the authors mean by ‘urban organisms’? Is this biodiversity in cities?
57 ‘…received extensive attention for…’ Do authors mean that the research is being applied in conservation practice?
68-78 This part combines two urban green space functions that are not clearly separated: positive effects (ecosystem functions) to sustain biodiversity, and ecosystem functions for the quality of life of humans in the city.
83 Why is it necessary or urgent to do a systematic review? How do you know an overview is lacking in academia or practice, and what knowledge exactly would be needed?
85 What is the current problem behind the two questions (academic, societal)? Regarding question 1: progress is somewhat vague and not what you’re measuring. What do you want to analyze exactly, the development of research themes? Question 2: do you mean by hot spot the research focus?
89 Before going into data, more specific (sub)questions are needed, which can be answered by the data analysis. How does the method answer them?
98 The use of a bibliometric software package does not guarantee a systematic review. Explain why this package is chosen (what does it do to answer the questions). Systematic review is largely about gathering and treating the sources in a systematic way, documented step-by-step (see the example of the PRISMA method for systematic review).
104/105/111 What do these three methods do exactly?
111 To clarify the evolution, are frequencies analyzed over time?
114 What does ‘they’ refer to?
119 Result section lack introductory text, starting directly with a figure without explanation decreased the readability.
120 Outlier of (incomplete?) year 2022 seems to influence the trend line
123 Remarkable that China is has second position in production of research, but is cited a lot less. Is there an explanation?
130-131 Round off 2.9
134 Are these citations referring to any US articles or the analyzed set?
146 Are sources journals? Doc = article?
148 map is not discussed, research centres not clearly visible
153 Index = number of collaborations with other countries? Not discussed much in text.
162-172 Interesting analysis
173 b) how does this analysis work exactly, not explained in the methodology section, what do the axes represent?
176 Word cloud
188 Graph similar to Fig. 1, but cut-off at other year
189-200 Terminology of ‘node keywords’ and the different stages should be explained earlier, in the methodology section. Also the existence of two additional stages, which the topic apparently has not reached yet.
201 Interesting figure. Smallest dot in legend is larger than small dots in the graph. Why freq > 4?
207-208 Number of papers shows exponential growth from 2008.
212 What is urban construction? Planning?
214 Would it be possible to combine both timelines and discuss the information at once?
216 What do authors mean by ‘there is still a lot of research in the future?’, that a lot still has to be done? Different from stages of urban heat island effect research (how, slower?)
2017 Li et al 2012 could be discussed in method section to introduce the concepts of stages
220 Reason for what? Slower development?
223 ‘space’ = location?
224-234 Are there different developments in terms of numbers and themes in the four regions? That would make it more relevant for the main issue of the paper.
231 urban construction?
232 Are urban areas really bigger than rural areas?
234 Attention from whom? Is a research centre a case or a group of knowledge institutes?
239 Rocchi et al 2020 discuss in methodology section for bibliographic review?
244 Are ‘hot issues of one research’ the key topics in a research field?
260-263 I do not understand this
267-268 which questionnaire and simulations?
273 Font size. Hot spots = agenda?
274-316 This part is rather different from the above, it is not a result from the quantitative analysis, but rather a qualitative literature review regarding some (leading?) publications in the field. How does it relate to the method as described earlier? How do you select the literature to discuss – are these the top performing papers in a cluster? Are research gaps as described in the gathered papers analyzed? The word ‘should’ (282,285) suggests authors are proposing a research agenda, not analyzing the documents.
297 may be a hot issue in the future? How do you know?
311-312 idem
318 What was the issue of the paper. On what is the first statement of the conclusion based?
322-323 main themes
329-333 Where do these suggestions come from, qualitative literature review?
Author Response
Biodiversity in urban green areas is a relevant field of study, on which a systematic review might make an important contribution. For the submitted paper ‘Development and the sustainability of the urban green biodiversity: The current situation and future prospects’ to realize this potential, I believe several improvements are necessary.
We are grateful to you for the recognition. We have followed your recommendation, modified the introduction, method, discussion and conclusion, and added a new content at the end of the results.
Issue (1): The title does not seem to represent the approach of the article. Rather than analysing development practice and sustainability, it deals with topics and metrics of the research field. A suggestion: ‘Biodiversity in urban green areas: a bibliographic review on the current research field and its prospects’
Response: Thanks for your constructive suggestions. We think the title you gave is more consistent with the content of our article, and we have modified it and made minor adjustments.
Line 1-2: Biodiversity in urban green areas: a bibliometric review on the current research field and its prospects.
Issue (2): A convincing problem statement is lacking. Why is it needed to have a better overview of the research field and its prospects, in the academic context and/or in practice? The ‘gap’ statement in line 12 that a systematic review is lacking, does not convince the reader of the urgency, and one wonders whether there are not yet some good overviews available of literature of biodiversity in cities. The article itself already cites a meta-analysis by Beninde (in the latter part), which is a form of systematic review.
Response: We apologize for the lack of convincing problem statement and thank you for your question. To solve this problem, we added our statistics on review articles in this field, and explained the direction, which proved that this field is indeed lack of systematic review.
Line 104-108: In recent years, a lot of studies on urban green space diversity have emerged. There are 156 reviews in total, including 30 reviews focusing on cities, of which only one focuses on agriculture biodiversity of urban green space. As far as we have seen, systematic review based on bibliometrics has not been reported.
Issue (3): The described method is a statistical analysis of a list of publications, while especially the future prospects seem to be the result of a qualitative literature review, of which the method is not explained. Recommendation: either choose for the quantitative approach exclusively, or describe in more detail and in order the mixed approach. Why and how are the methods combined?
Response: Thanks for your constructive suggestions. We added quantitative analysis in the last part of the results to explain the future research focus proposed in 4.3 of our discussion.
Line 283-288: The blue part is in the first and second quadrants. Vegetation, climate change, dynamics and other topics are well developed in this part. The red part is in the first quadrant. Urban, health, benefit and other topics have developed well and are closely related to other topics. In the first and fourth quadrants, the purple part including topics such as pattern, community, ecology is closely related to other topics. Green and orange parts are in the third quadrant, and the themes of metaanalysis and heterogeneity develop slowly and are relatively isolated (Fig.8).
Line 289-294: Fig. 8 Thematic map. The horizontal axis is the centrality, which represents the relevance degree between a certain theme and other themes. It can be interpreted as the importance of this theme in the entire field development. The vertical axis is the density, which represents the development degree within a certain theme. It can be interpreted as the development status of the theme.
Issue (4): The article contains (too) many figures, several of which are not explicitly used in the text to make a point. The purpose of several figures remains unclear, for a lack of specific research questions. For example: location information is used to affirm the existence of 4 research centers, but what does that contribute to the main issue of knowing what current and future themes are? The focus of the paper seems to be development of themes over time, so why not select figures to emphasize that? Response: Thanks for your constructive suggestions. We illustrate the changes of scientific and technological achievements published by countries around the world through time changes, so that we can clearly see the changes of scientific research development and research centers in countries around the world.
Line 207-211: Between 1998 and 2008, there was only a small part of the cooperation and exchanges between Europe and the United States (Fig.3a). Between 2009 and 2021, Europe, the United States, China, and Australia had the closest academic exchanges, among which the United States had the highest number of cooperation (464), followed by the United Kingdom (350) and Germany (269) (Fig.3b).
Line 213-219: Fig. 3 Country Collaboration Map. The red lines represent the cooperation and exchanges between the authors of the two countries. The more lines, the closer the exchanges. The numbers on the right represents the times a country exchanges scientific products with other countries. (a) 1998 to 2008. (b) 2009 to 2021.
More detailed remarks per line in the manuscript:
15/20 If 2008 is the changing point in the development of the field, why separate budding and development stages in the year 2012 and not in 2008?
Response: Thanks for your question. In 2008, we divided it according to the number of articles issued, while in 2012, we divided it according to the keywords. The meanings of the two are different. The former emphasizes the change in quantity, while the latter emphasizes the development of the theme.
38 Why ‘therefore’? The phrase means ‘urban green space has gradually become a hot spot for urban biodiversity’, correct? How does this follow from the above?
Response: Thanks for your question. We have revised our statement. The hot spots need more specific quantitative analysis. It can only be concluded from the above that urban green space is very important and a focus.
Line 55-56: Therefore, urban green space has gradually become an important field of urban biodiversity research.
41 Why is ‘walled or not’ relevant?
Response: Thanks for your question. We removed this expression to make it clearer.
Line 59: It was defined as any land without buildings.
42/45 references in caps, unlike rest of article.
Response: We apologize for the mistake. We have modified it in line 60 and 63.
48 What is the function of public green space? What is production protection land?
Response: Thanks for your question. We have added the introduction of two green spaces.
Line 65-67: The former is mainly used to arrange recreational facilities for residents to share, while the latter has the functions of sanitation, isolation and safety protection.
53 Which ecosystem functions, and why?
Response: Thanks for your question. We've already given examples.
Line 73-74: such as improving air quality, reducing noise, repairing polluted soil, providing food and raw materials for residents, etc.
55 What do the authors mean by ‘urban organisms’? Is this biodiversity in cities?
Response: We apologize for the statement mistake. We have revised the statement.
Line 75: In addition, urban green space has been proved to be a refuge for biodiversity.
57 ‘…received extensive attention for…’ Do authors mean that the research is being applied in conservation practice?
Response: We apologize for the statement mistake. We have revised the statement.
Line 76-77: Many studies have also discussed the urban biodiversity pattern, so as to provide better suggestions for biodiversity conservation.
68-78 This part combines two urban green space functions that are not clearly separated: positive effects (ecosystem functions) to sustain biodiversity, and ecosystem functions for the quality of life of humans in the city.
Response: Thanks for your question. We have divided them into two paragraphs to describe their functions.
Line 88-101: Scholars have also widely discussed the ecosystem function of urban green space biodiversity (Gallo et al., 2017). The biodiversity of urban green space significantly affects the green space microenvironment, such as soil carbon storage (Guillen-Cruz et al., 2021), soil temperature (Jenerette et al., 2011), soil moisture (Guillen-Cruz et al., 2021), green space biomass (Lahoti et al., 2020), etc. Urban green space also plays a positive role in improving urban ecosystem functions, such as mitigating the heat island effect (Yuan et al., 2021a), and regulating water runoff (Dhakal and Chevalier, 2017), etc. In addition, urban green space can also effectively deal with the adverse effects and risks of climate change, and reduce the disaster losses caused by extreme weather and climate events.
In addition, many studies have confirmed that urban green space plays a key role in improving human well-being, such as relieving stress and fatigue (Fan et al., 2011), reducing noise pollution (Kogan et al., 2021), reducing disease occurrence (Seo et al., 2019), reducing crime rate (Shepley et al., 2019), and improving education quality (Wolsink, 2016). It has greatly improved the living quality of residents and their physical and mental health.
83 Why is it necessary or urgent to do a systematic review? How do you know an overview is lacking in academia or practice, and what knowledge exactly would be needed?
Response: Thanks for your question. We added our statistics on review articles in this field, and explained the direction, which proved that this field is indeed lack of systematic review.
Line 104-106: In recent years, a lot of studies on urban green space diversity have emerged. There are 156 reviews in total, including 30 reviews focusing on cities, of which only one focuses on agriculture biodiversity of urban green space.
85 What is the current problem behind the two questions (academic, societal)? Regarding question 1: progress is somewhat vague and not what you’re measuring. What do you want to analyze exactly, the development of research themes? Question 2: do you mean by hot spot the research focus?
Response: Thanks for your question. In question 1, we refined the questions we raised, and in question 2, we modified the statement. We think your question is very good, and the statement of research focus should be more appropriate than the hot spot.
Line 110-113: 1) How about the research progress of global urban green space biodiversity in the past few decades, including the number of articles, the different stage, the differences and cooperation between countries? 2) What is the research focus of urban green space biodiversity in the future?
89 Before going into data, more specific (sub)questions are needed, which can be answered by the data analysis. How does the method answer them?
Response: Thanks for your constructive suggestions. We have refined our research questions based on the previous comment and pointed out our aim.
98 The use of a bibliometric software package does not guarantee a systematic review. Explain why this package is chosen (what does it do to answer the questions). Systematic review is largely about gathering and treating the sources in a systematic way, documented step-by-step (see the example of the PRISMA method for systematic review).
Response: Thanks for your constructive suggestions. We pointed out the advantages of the software packages we used and described its main functions.
Line 126-129: It has a relatively complete bibliometric analysis process of data import, transformation, data analysis and scientific visualization, including two series of functions: 1) bibliometric basic analysis and analysis index extraction; 2) mining of literature related concepts, knowledge and social structure.
104/105/111 What do these three methods do exactly?
Response: Thanks for your question. We wrote the methods separately and explained in detail what each method did.
Line 134-136: We analyzed the basic information of the 3806 articles. In order to explore the development of the field, we tested the changing trend through Mann Kendall. In order to explore the significant changes in a certain year, we used Pettitt method to test the change point (Conte et al., 2019).
Line 142-144: In addition, in order to clarify the evolution of research topics in this field, word cloud was firstly extracted through wordcloud2 function package to identify the frequency of keywords and find the key issues.
111 To clarify the evolution, are frequencies analyzed over time?
Response: Thanks for your constructive suggestions. We added the analysis of frequency over time to the results.
Line 267-268: Before 2012, the frequency of keywords is relatively low.
Line 271-272: Since 2012, the frequency of keywords has increased significantly, and some new keywords have appeared.
114 What does ‘they’ refer to?
Response: We apologize for the lack of clarity. We have revised the expression.
Line 170: Based on co-occurrence keywords, research field can be used to divide different research stages.
119 Result section lack introductory text, starting directly with a figure without explanation decreased the readability.
Response: Thanks for your constructive suggestions. We have added introductory text before each result.
Line 189-194: Descriptive analysis mainly involves indicators such as articles, keywords, authors, institutions, cited times, published years, etc., which are mainly divided into performance and evaluation indicators. Performance indicators describe intuitive phenomena, such as the number of articles, the number of citations, etc. Evaluation indicators are intended to quantitatively evaluate the scientific contributions of articles, authors, magazines, institutions, countries, etc., such as collaboration index, ranking of published articles, etc.
Line 228-232: The citation of scientific literature shows the inheritance and utilization of scientific knowledge, as well as the connection and development between events in the process of scientific development. Co-citation analysis aims to compare, analyze and cluster the phenomenon of co-citation in scientific papers by using library science and statistics.
Line 253-254: Thematic analysis clusters different themes, points out focus, shows the evolution process of themes over time, and forecasts the development trend of future themes.
120 Outlier of (incomplete?) year 2022 seems to influence the trend line
Response: We apologize for the mistake. 2022 refers to some articles published online in advance. We have eliminated the data of 2022 in the follow-up of the whole study. When making the trend chart, the year 2022 appears in the chart due to negligence. I'm sorry. We have redrawn the figure and obtained new change point and trend line, which have no impact on the analysis of this study. New figure in line 180.
123 Remarkable that China is has second position in production of research, but is cited a lot less. Is there an explanation?
Response: Thanks for your question. We have explained in the discussion section.
Line 330-334: Although China's scientific research achievements rank second, the number of citations is small. Considering that China has only gradually developed in recent years, it takes a certain time for the accumulation of scientific research achievements. In addition, Chinese scholars pay more attention to regional studies.
130-131 Round off 2.9
Response: Thanks for your constructive suggestions. We have modified it.
Line 197: The average number of citations of 3806 articles per year is 2.9.
134 Are these citations referring to any US articles or the analyzed set?
Response: Thanks for your question. These citations refer to all American articles retrieved through our method.
146 Are sources journals? Doc = article?
Response: We apologize for the mistake. We have modified it in line 212.
148 map is not discussed, research centres not clearly visible
Response: Thanks for your constructive suggestions. We made two figures according to the change point, showing the changes in the number of articles in the two periods in different regions, which were also explained in the discussion. New figure in line 213.
153 Index = number of collaborations with other countries? Not discussed much in text.
Response: Thanks for your question. We put the original index in the picture and explained it in the legend. In line 213.
162-172 Interesting analysis
Response: Thank you for your strong interest in our analysis.
173 b) how does this analysis work exactly, not explained in the methodology section, what do the axes represent?
Response: Thanks for your constructive suggestions. We have explained it specifically in the method section.
Line 152-158: MCA is a method that allows studying the association between two or more qualitative variables.MCA can also be understood as a generalization of Correspondence Analysis (CA) to the case where there are more than two variables.A series of transformations allows the computing of the coordinates of the categories of the qualitative variables, as well as the coordinates of the observations in a representation space that is optimal for a criterion based on inertia. The percentage of adjusted inertia that corresponds to each axis and the percentage of adjusted inertia cumulated over the two axes are displayed on the map.
176 Word cloud
Response: Thank you for pointing out my mistakes. We have modified it.
Line 249: The word cloud which has 27 keywords with frequency over 50.
188 Graph similar to Fig. 1, but cut-off at other year
Response: We apologize for the mistake. We have modified Figure 1 to 2021. In line 180.
189-200 Terminology of ‘node keywords’ and the different stages should be explained earlier, in the methodology section. Also the existence of two additional stages, which the topic apparently has not reached yet.
Response: Thanks for your constructive suggestions. We have explained the node keywords in the method section. In addition, we agree with you that our research topic has not reached the last two stages.
Line 174-175: We extracted co-occurrence keywords, and extracted the top 50% co-occurrence keywords as node keywords, and judged the development stage based on the trend.
201 Interesting figure. Smallest dot in legend is larger than small dots in the graph. Why freq > 4?
Response: Thanks for your question. We have modified the fig7, deleted the legend, and added specific numbers after each line. In addition, the reason why the frequency is greater than 4 is that we find that when the frequency of keywords is greater than 4, the change of keywords is very small. On the contrary, when the frequency is less than 4, the change of keywords is very large. We think such a decision is more persuasive for judging the evolution of the theme.Fig7 in line 278.
207-208 Number of papers shows exponential growth from 2008
Response: Thanks for your constructive suggestions. We have revised the statement in line 297.
212 What is urban construction? Planning?
Response: Thank you for pointing out my mistakes. We have modified it in line 303.
214 Would it be possible to combine both timelines and discuss the information at once?
Response: Thanks for your constructive suggestions. We have carefully considered your suggestion, but since the two time points represent different meanings, we think it is difficult to discuss them together. We have also explained in the above reply.
216 What do authors mean by ‘there is still a lot of research in the future?’, that a lot still has to be done? Different from stages of urban heat island effect research (how, slower?)
Response: We apologize for the lack of clarity. We have revised the statement.
Line 307-309: At present, it is still in the development stage, and a lot still has to be done in the future. The research is developing more slowly than the research in urban heat island effect.
217 Li et al 2012 could be discussed in method section to introduce the concepts of stages
Response: Thanks for your constructive suggestions. We introduced the concepts in the method.
Line 170-172: Based on co-occurrence keywords, research field can be used to divide different research stages, generally including four stages: budding stage, development stage, lull stage and maturation stage (Li et al., 2021).
220 Reason for what? Slower development?
Response: We apologize for the lack of clarity. We have revised the statement.
Line 312-314: The reason for the slower development of this research field may be that the surface temperature in the city is more easily perceived by residents and more closely related to human life.
223 ‘space’ = location?
Response: Thanks for your question. We have modified it in line 318.
224-234 Are there different developments in terms of numbers and themes in the four regions? That would make it more relevant for the main issue of the paper.
Response: Thanks for your constructive suggestions. In the results, we have mapped the development of the number of articles published in different regions. In addition, the theme development in different regions needs more detailed analysis. Our research emphasizes the development of the theme itself. Thank you again for your good suggestions.
231 urban construction?
Response: Thank you for pointing out my mistakes. We have modified it in line 326.
232 Are urban areas really bigger than rural areas?
Response: We apologize for the mistake. We have revised the statement.
Line 326-327: The proportion of population in urban areas has exceeded that in rural areas, entering the development period of big cities.
234 Attention from whom? Is a research centre a case or a group of knowledge institutes?
Response: We apologize for the lack of clarity. We have revised the statement.
Line 329-330: Therefore, it has attracted extensive attention of global scholars, especially Chinese scholars, and has become a research center with many knowledge institutions.
239 Rocchi et al 2020 discuss in methodology section for bibliographic review?
Response: Thanks for your question. We have supplemented the statement.
Line 338-340: They also conducted keyword clustering and theme trend research, and finally found that the keywords in the later stage tend to be related with human well-being.
244 Are ‘hot issues of one research’ the key topics in a research field?
Response: Thanks for your question. Our statement is not accurate and has been revised.
Line 345: The key topics of one research will vary greatly in different periods.
260-263 I do not understand this.
Response: We apologize for the lack of clarity. We have revised the statement.
Line 361-364: Control experiments and mathematical modeling are the main methods (Xie et al., 2012). The theme of the “Ecological Function of Urban Green Biodiversity” focuses on the level of urban functionality.
267-268 which questionnaire and simulations?
Response: Thanks for your question. We added the specific questionnaires and simulations.
Line 369-370: such as questionnaire on residents' sports activities or physical and mental health, and scenario simulation of temperature, climate and other factors,
273 Font size. Hot spots = agenda?
Response: We apologize for the mistake. We have modified the font size and replaced the word "hot spot" with the more accurate word “focus”.
Line 376: 4.3 Future research focus of urban green space biodiversity
274-316 This part is rather different from the above, it is not a result from the quantitative analysis, but rather a qualitative literature review regarding some (leading?) publications in the field. How does it relate to the method as described earlier? How do you select the literature to discuss – are these the top performing papers in a cluster? Are research gaps as described in the gathered papers analyzed? The word ‘should’ (282,285) suggests authors are proposing a research agenda, not analyzing the documents.
Response: Thanks for your question. First of all, we have added a new quantitative analysis method and explained it in the method. Secondly, the literature we discussed in the discussion is the literature with a high citation rate in this category, which is very persuasive. Finally, the word "gap" in the discussion is not accurate, and the expression has been modified. In addition, the word "should" has also been modified in the article, I am very sorry for the inaccuracy of our statement before, and thank you again for your suggestions.
Line 390-391: Therefore, in the future, we need to invest more efforts in the study of microbial groups in urban green space (Fig. 8).
Line 412-413: However, there are relatively few studies on urban green space irrigation water and residential water,
297 may be a hot issue in the future? How do you know?
Response: Thanks for your question. We have changed the word "hot spot" to "focus", and the new quantitative analysis method has been added to explain the future key topics in the discussion.
311-312 idem
Response: idem
318 What was the issue of the paper. On what is the first statement of the conclusion based?
Response: Thanks for your question. Our issue is to study the prospect of urban green space biodiversity. Therefore, we revised the first sentence of the conclusion to emphasize the importance of urban green space biodiversity and the purpose of our research.
Line 424-426: Biodiversity of urban green space plays a huge role in promoting human welfare and achieving sustainable urban development. We reviewed the field of urban green space biodiversity, hoping to help scholars from all countries understand its theme and prospects.
322-323 main themes
Response: Thanks for your constructive suggestions. We added this expression to the text.
Line 429: There are three main themes in this field.
329-333 Where do these suggestions come from, qualitative literature review?
Response: Thanks for your question. We have rewritten it and based on the evidence.
Line 435-440: We should invest more effort in rational planning according to the construction strategies of different cities. Besides, according to the 17 sustainable development goals proposed by the United Nations in 2015, we should solve ecological problems in the future in combination with society and economy, so as to protect global biodiversity, improve the quality of urban ecosystems, improve the health and well-being of urban residents, and finally move towards a sustainable path.

Round 2
Reviewer 2 Report
The authors have considerably improved the article and given detailed feedback on the comments in round 1. I recommend publication after an English language check and typos such as beginning of line 167.